# Geographically persistent clusters of La Crosse virus disease in the Appalachian region of the United States from 2003 to 2021

**Corey Allen Day** [1]*, **Agricola Odoi**[2], **Rebecca Trout Fryxell**[1]

**1** Department of Entomology and Plant Pathology, University of Tennessee, Knoxville, Tennessee, United States of America, **2** Department of Biomedical and Diagnostic Sciences, University of Tennessee, Knoxville, Tennessee, United States of America

* cday11@vols.utk.edu

**Data Availability Statement:** The case data underlying figures and analyses were uploaded to Dryad with permission from the Centers for Disease Control and Prevention Division of Vector-

## Abstract

La Crosse virus (LACV) is a mosquito-borne pathogen that causes more pediatric neuroinvasive disease than any other arbovirus in the United States. The geographic focus of reported LACV neuroinvasive disease (LACV-ND) expanded from the Midwest into Appalachia in the 1990s, and most cases have been reported from a few high-risk foci since then. Here, we used publicly available human disease data to investigate changes in the distribution of geographic LACV-ND clusters between 2003 and 2021 and to investigate socioeconomic and demographic predictors of county-level disease risk in states with persistent clusters. We used spatial scan statistics to identify high-risk clusters from 2003–2021 and a generalized linear mixed model to identify socioeconomic and demographic predictors of disease risk. The distribution of LACV-ND clusters was consistent during the study period, with an intermittent cluster in the upper Midwest and three persistent clusters in Appalachia that included counties in east Tennessee / western North Carolina, West Virginia, and Ohio. In those states, county-level cumulative incidence was higher when more of the population was white and when median household income was lower. Public health officials should target efforts to reduce LACV-ND incidence in areas with consistent high risks.

## Author summary

La Crosse virus (LACV) is a mosquito-borne pathogen that can cause neuroinvasive disease (e.g., encephalitis and meningitis), usually in children. In the 1990s, the geographic focus of reported LACV neuroinvasive disease (LACV-ND) shifted from the Midwest into the Appalachian region, and most cases have been reported from a few areas in those regions since then. Here, we used publicly available human disease data to investigate changes in geographic clusters of LACV-ND from 2003–2021 and to identify socioeconomic and demographic variables that are associated with disease risk. We found a consistent distribution of areas with elevated disease risk (i.e., clusters), with three persistent clusters in Appalachia and one intermittent cluster in the upper Midwest. In states with persistent clusters, county-level cumulative incidence increased when more of the

Borne Diseases (CDC DVBD), and can be obtained at https://doi.org/10.5061/dryad.djh9w0w31.

**Funding:** The author(s) received no specific funding for this work.

**Competing interests:** The authors have declared that no competing interests exist.

population was white and when median household income was lower. Public health officials should work to reduce LACV-ND incidence in the areas where disease risk is consistently high.

## Introduction

La Crosse virus (LACV), a California serogroup virus in the family Peribunyaviridae, is the leading cause of arboviral neuroinvasive disease among children in the United States (US) [1]. The natural LACV transmission cycle involves transmission within populations of the eastern tree hole mosquito, *Aedes triseriatus* (Diptera: Culicidae), and transmission between *Ae. triseriatus* and small mammal amplifying hosts [2–4]. Human LACV infections have been reported in the eastern US since the virus was initially isolated in 1964 [1,5–7], but the total incidence is thought to be severely underestimated [8]. Most infections are asymptomatic or cause febrile illness, but rarely, severe LACV infections can become neuroinvasive [9]. La Crosse virus neuroinvasive disease (LACV-ND) typically presents as pediatric encephalitis, meningitis, or meningoencephalitis, and often causes long-term neurologic sequelae such as reduced mental function and chronic seizure disorders [9–11].

Historically, most cases of LACV-ND in the US were reported from the Midwest (i.e., Illinois, Iowa, Indiana, Minnesota, Ohio, and Wisconsin) [6], but during the 1990s, the disease emerged in the Appalachian region [12]. By the early 2000s, the majority of LACV-ND cases were reported from West Virginia, east Tennessee, western North Carolina, and south-central Ohio [13]. La Crosse virus neuroinvasive disease has not emerged in new areas since then, indicating that the distribution of disease is presently stable, with counties in a few high-risk foci continuously reporting the majority of LACV-ND cases in the United States [5].

If the geographic distribution of LACV-ND is consistent, public health officials can efficiently reduce disease incidence by implementing interventions in areas with persistent risk. Here, we use 24 years of reported human LACV-ND case data from the national arboviral disease surveillance system, ArboNET [14], to investigate the persistence of geographic LACV-ND clusters and to identify socioeconomic and demographic predictors of LACV-ND risk in states with persistent clusters. The results of this study will elucidate spatial-temporal patterns of LACV-ND clustering and guide resource allocation to reduce the burden of LACV-ND.

## Methods

### Ethical statement

The University of Tennessee Institutional Review Board (IRB Number: UTK IRB-22-07077-XM) determined that IRB review was not required because research did not use data that meets the definition of protected health information.

### Study area and period

This study was conducted in the eastern US, where the vast majority of LACV-ND is reported [5]. The study area consisted of 31 states, 2,046 counties, and the entire Appalachian region (Fig 1). We used the US Census Bureau 2020 TIGER/Line shapefiles to define state and county boundaries [15] and used the Appalachian Regional Commission's definition of Appalachia to define Appalachian counties [16]. To compare the geographic distribution of LACV-ND over

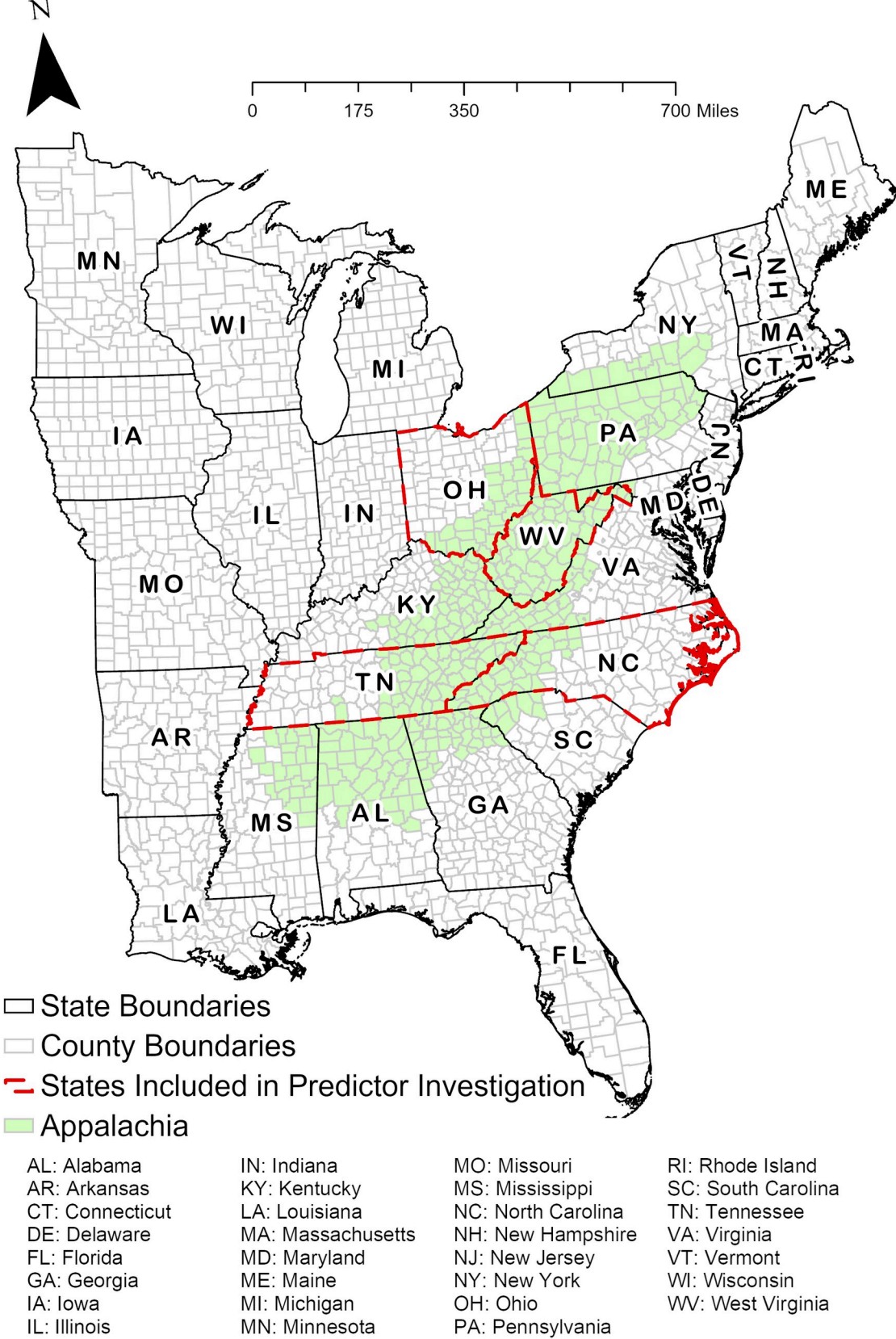

**Fig 1. Map of the study area (eastern United States) with state boundaries, county boundaries, boundaries of states used for investigation of risk predictors, and state abbreviations.**

the entire study period (2003–2021), data were grouped into the following four time periods for analyses: 2003–2007, 2008–2012, 2013–2017, and 2018–2021. For the investigation of county-level predictors of disease risk, we only included the states that contained geographic disease clusters in every time period (Fig 1) (see Methods: Geographic Cluster Analysis for a definition of disease cluster).

## Case definition

Because of its status as a nationally notifiable disease, states report both probable and confirmed cases of LACV-ND. An arboviral infection is reported as neuroinvasive when an individual presents with encephalitis, meningitis, acute flaccid paralysis, or other disfunctions of the central nervous system [17]. A confirmed case of LACV-ND must meet one or more of the following laboratory criteria: isolation of LACV from tissue or demonstration of specific LACV antigen or nucleic acid in tissue or body fluid; a four-fold or greater change in LACV-specific antibody titers in paired sera; LACV-specific IgM antibodies in cerebrospinal fluid and a negative result for other IgM antibodies for arboviruses endemic to the geographic region of exposure [17]. A case is reported as probable when there is laboratory evidence of LACV-specific IgM antibodies in cerebrospinal fluid but no additional testing [17].

## Case data

We obtained all probable and confirmed cases of LACV-ND reported in the US from 2003–2021 from the US Centers for Disease Control and Prevention national surveillance system for arboviral diseases, ArboNET. The data were provided at the county level and only included the annual number of cases per county. ArboNET does not provide case-specific demographic information (e.g., age, sex) or epidemiolocal information (e.g., date confirmed) at the county level.

## County level demographic and socioeconomic data

We used the National Historical Geographic Information System (NHGIS) [18] to obtain county-level demographic and socioeconomic data from period-specific US census surveys (Table 1). Most data came from 5-year American Community Survey (ACS) or decennial census data, but we also obtained county-level population estimates for 2005 from the NHGIS 1970–2007 Vital Statistics: Natality & Mortality dataset, which derived data from the US Census Bureau USA Counties database [19].

## Descriptive analyses

We calculated the number of reported LACV-ND cases within the study area and identified the five states that reported the most cases from 2003–2021. To describe the focal reporting of LACV-ND within those states, we calculated the proportion of counties that reported at least

**Table 1. Time period-specific data sources for county-level total population and potential predictor variables.**

| Time Period | Data Source (Total Population) | Data Source (Potential Predictors) |
|---|---|---|
| 2003–2007 | 1997–2007 VS[1] | 2009 5-year ACS[2] |
| 2008–2012 | 2010 Decennial Census | 2010 Decennial Census |
| 2013–2017 | 2015 5-year ACS | 2015 5-year ACS |
| 2018–2021 | 2019 5-year ACS | 2019 5-year ACS |

[1]Vital Statistics: Natality & Mortality Data
[2]ACS: American Community Survey

one case during the study period. For each time period, we calculated the crude cumulative incidence (CI) (i.e., the number of cases in an area divided by the total population in that area) for the entire study area, for the five states that reported the most cases during the entire study period, and for each county. We visualized the county-level CIs as choropleth maps using Jenk's classification scheme.

## Geographic cluster analysis

We used geographic cluster analyses to test the null hypothesis that reported cases of LACV-ND were distributed randomly throughout geographic space. The null hypothesis of spatial randomness is rejected when a group of neighboring areas (i.e., cluster) has a significantly higher CI than the overall study area. We used Tango's restricted flexibly-shaped scan statistic (FSSS), which scans the study area for potential clusters up to a user-specified maximum size [20,21]. Our cluster analysis was conducted at the county level for each time period. We used the R package spdep version 1.2.2. [22] to calculate queen spatial weights, which describe neighbor relationships among the counties. We identified clusters for each time period using Tango's restricted FSSS [20] implemented with the R package rflexscan version 1.0.0 [23], specifying a Poisson model with a maximum cluster size of 40 counties and significance level of 0.01.

To compare LACV-ND risk within the clusters to risk in the study area overall, we computed the CI and risk ratio (RR) (i.e., the CI within the cluster divided by the CI in the entire study area) for each cluster. Finally, to identify counties that were consistently present in clusters, we calculated the number of time periods that each county appeared in a significant cluster (minimum = 0 time periods, maximum = 4 time periods) and presented the results as a classed choropleth map.

## Predictor investigation

Because most states in the study area reported few or no cases, we chose to limit our investigation of risk predictors to the four states that contained high-risk clusters in every time period (Figs 1 and 3) to understand risk factors in the states that consistently reported cases. We chose the following variables as potential predictors based on existing evidence of their association with LACV-ND [5,24,25]: percentage of the population aged 20 years or younger, percentage of the population with a high school diploma or less, median household income, median year that households were built, population density per square kilometer, housing density per square kilometer, and percentage of the population that is white. To avoid problems with multicollinearity, we calculated univariable correlations between the potential predictors using Spearman rank correlations. Only one variable from a pair of highly correlated ($|r_s| >$ 0.7) potential predictors was retained for further assessment.

For model fitting we used the R package glmmTMB version 1.1.2.3 [26]. We initially used a Poisson distribution, but because overdispersion was consistently detected, we opted to use a negative binomial distribution with a log-link function. In all models, the dependent variable was the number of LACV-ND cases per county, grouped by time period, with the natural log of the total county population as the offset. We chose to include each time period separately in the models because it allowed us to account for temporal changes in the predictor variables (e.g., increasing housing density); this decision meant that each county is included in the model four times (once for each time period), which likely produced clustered data. To address the potential clustering, we included county and time period as random effects.

Prior to fitting multivariable models, we fit univariable models of disease risk with each potential predictor using a critical p-value of 0.10. We included variables that were significantly associated with disease risk in manual backwards elimination with a significance level of

**Table 2. Cumulative incidence per 100,000 population of La Crosse virus neuroinvasive disease for the eastern United States (US) and the five states that reported the most cases from 2003–2021.**

| Time period | Eastern US | West Virginia | North Carolina | Ohio | Tennessee | Wisconsin |
|---|---|---|---|---|---|---|
| 2003–2007 | 0.21 | 5.3 | 1.0 | 0.7 | 0.9 | 0.4 |
| 2008–2012 | 0.17 | 3.4 | 1.0 | 0.8 | 0.7 | 0.1 |
| 2013–2017 | 0.15 | 1.2 | 0.8 | 0.8 | 0.9 | 0.3 |
| 2018–2021 | 0.10 | 0.8 | 0.5 | 0.7 | 0.6 | 0.09 |

p < 0.05 to identify the most parsimonious model. To assess confounding, we calculated the change in model coefficients before and after eliminating each variable; if a predictor's coefficient changed by >20%, we considered the eliminated variable a confounder and included it in the final model. Once we identified the final model, we assessed spatial autocorrelation in the residuals with Moran's I and 9,999 Monte Carlo permutations for significance testing. We computed the RR of each predictor variable by exponentiating their regression coefficients.

## Results

### Descriptive analyses

From 2003–2021, a total of 1,261 cases of LACV-ND were reported in the study area with an average of 66.37 cases reported annually. Five states reported 87% of cases from 2003–2021: Ohio (OH; 346 cases), North Carolina (NC; 313 cases), West Virginia (WV; 195 cases), Tennessee (TN; 194 cases), and Wisconsin (WI; 52 cases). Across the four time periods, statewide CI in those states ranged from 0.1 to 5.3 cases per 100,000 population, while the CI in the total study area ranged from 0.1 to 0.21 cases per 100,000 population (Table 2).

West Virginia had the highest statewide CI in the study area during each time period, although it declined over time. From 2003–2007, the CI in WV was more than 25-times higher than in the eastern US overall, and from 2018–2021 it was 7.5-times higher (Table 2). North Carolina, OH, and TN each had the second-highest CI in at least one time period (Table 2). Notably, NC reported the highest maximum county CI in each study period (Fig 2), but the small proportion of NC counties that reported cases deflated the statewide CI.

Within the five states that reported the most cases, the proportion of counties that reported a case varied widely. Ohio had the highest proportion of counties that reported a case (76%), followed by WV (46%), TN (35%), NC (34%), and WI (28%). Visual assessment of CI indicated that LACV-ND risk was relatively high in western NC, east TN, WV, and OH throughout the entire study period (Fig 2). County-level CI was relatively high in southwest WI from 2003–2007 and 2013–2017, but not in the other time periods (Fig 2).

### Geographic cluster analysis

The number, size, and shape of statistically significant high-risk clusters varied across the time periods, but their general geographic distribution was consistent over time (Fig 3). At least three clusters were consistently present in the Appalachian region and one cluster was intermittently present in the Upper Midwest (i.e., Wisconsin, Iowa, and Illinois) (Fig 3). The primary cluster always included counties in western NC and east TN, while the second and third clusters included counties in either WV or OH. A single high-risk cluster was identified in the Upper Midwest from 2003–2007 and 2013–2017, but that region had no clusters in the other time periods. Kentucky, South Carolina, and Virginia each had one county in a cluster during at least one time period.

The clusters always contained the majority of cases from the overall study area (69% of cases from 2003–2007, 71% of cases from 2008–2012, 71% of cases from 2013–2017, and 76%

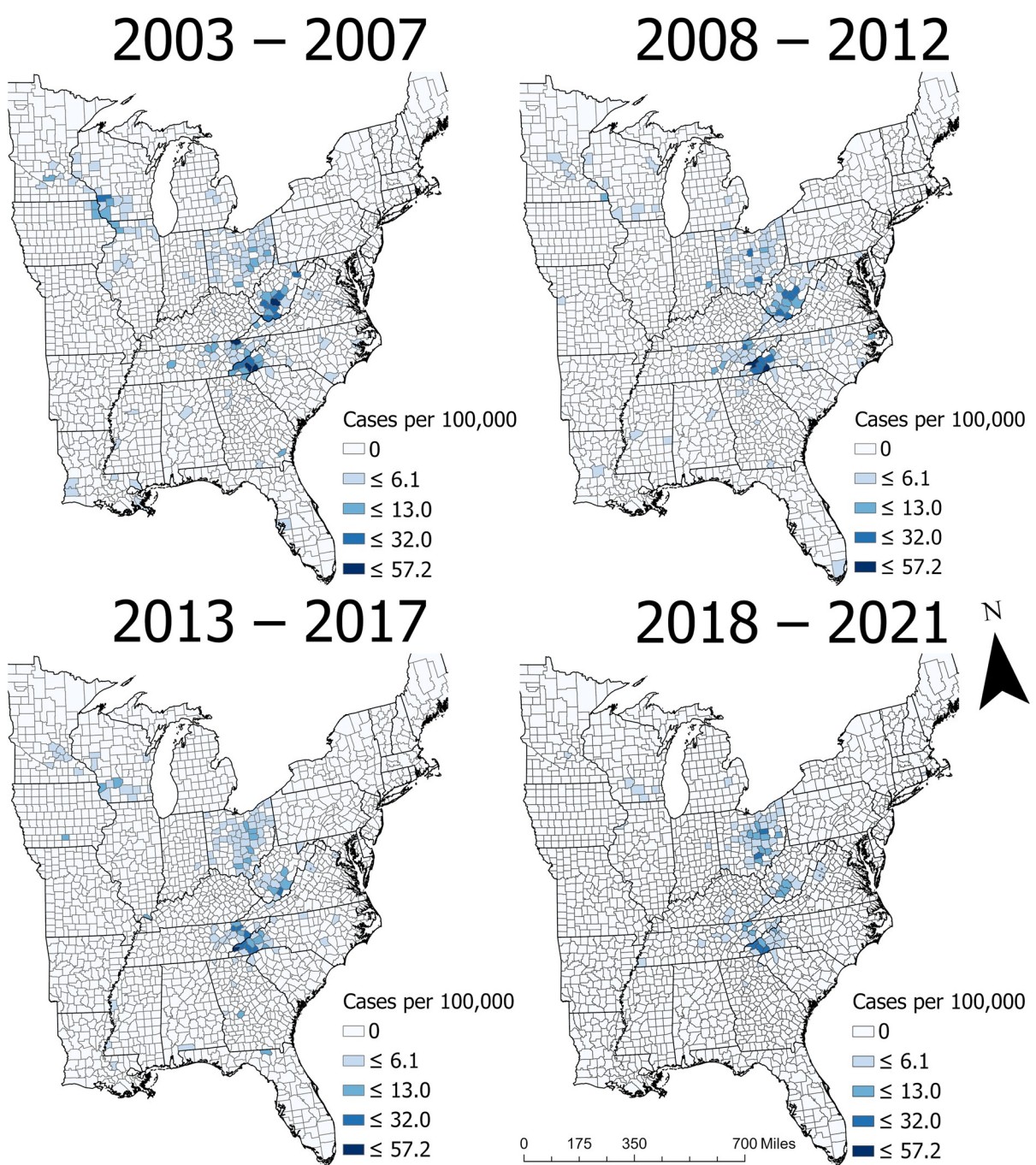

**Fig 2. Crude county-level cumulative incidence of La Crosse virus neuroinvasive disease in the eastern United States from 2003–2021 (data obtained from ArboNET).**

of cases from 2018–2021). Across all four time periods, 97 counties fell within a high-risk cluster at least once, and 74% of those counties were in the Appalachian region (Fig 4). Among the counties in a cluster, 23% were in a high-risk cluster in every time period (Fig 4).

Cumulative incidence within the clusters ranged from 18.02- to 70.33-times higher than in the study area overall (Table 3, Fig 3). The cluster in West Virginia had the highest RR from

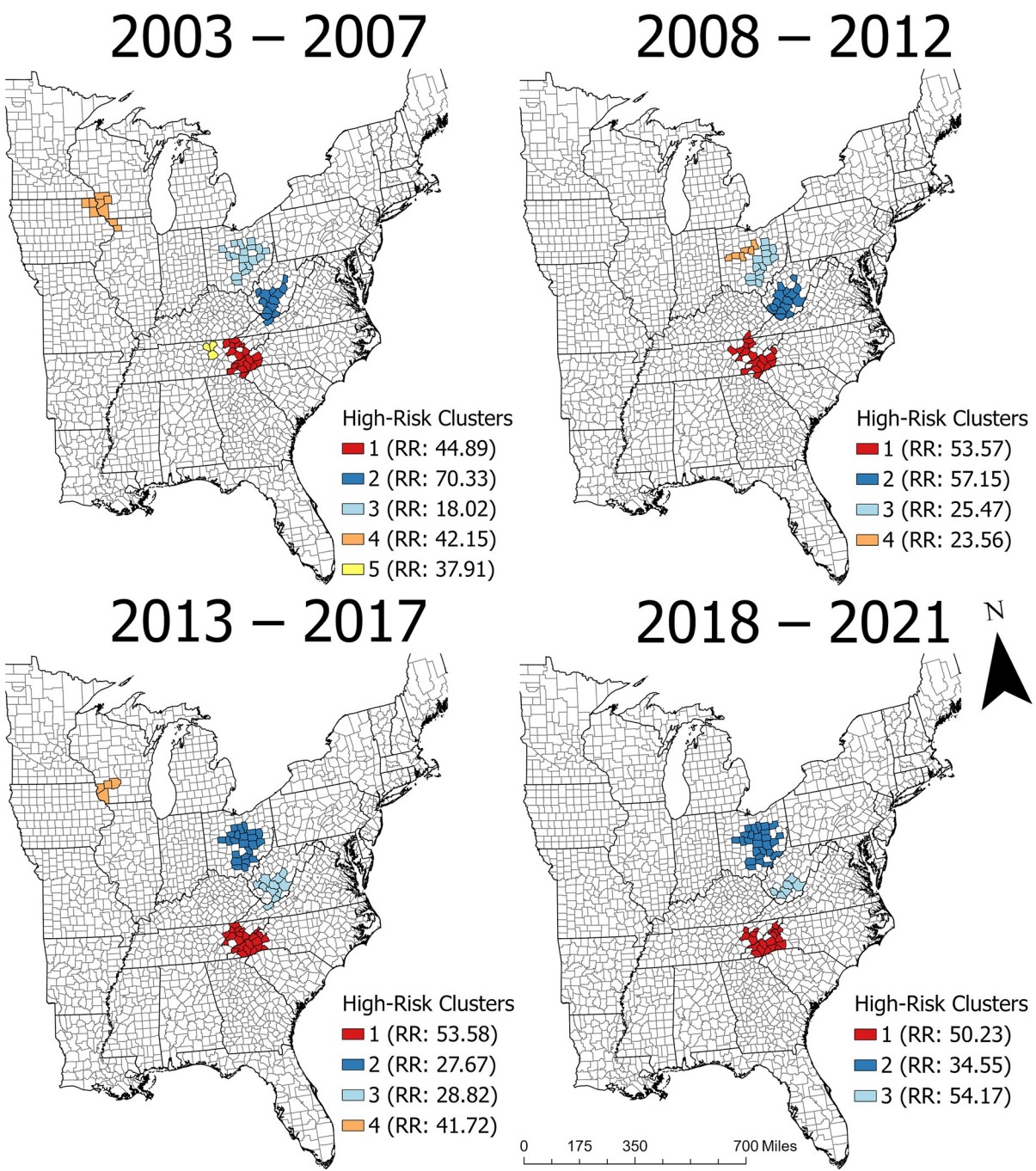

**Fig 3. Statistically significant high-risk clusters with risk ratios (RR) of La Crosse virus neuroinvasive disease in the eastern United States from 2003–2021 (data obtained from ArboNET).**

2003–2007 and 2008–2012, while the cluster spanning western North Carolina and east Tennessee had the highest RR from 2013–2017 and 2018–2021 (Table 3, Fig 3). Because the clusters included the majority of cases reported in each state, the CI within each cluster (Table 3) was often substantially higher than the relevant statewide CI (Table 2). For example, from 2013–2017, North Carolina and Tennessee had a combined CI of 0.8 cases per 100,000 population; during the same period, the CI in the high-risk cluster spanning east TN and western NC

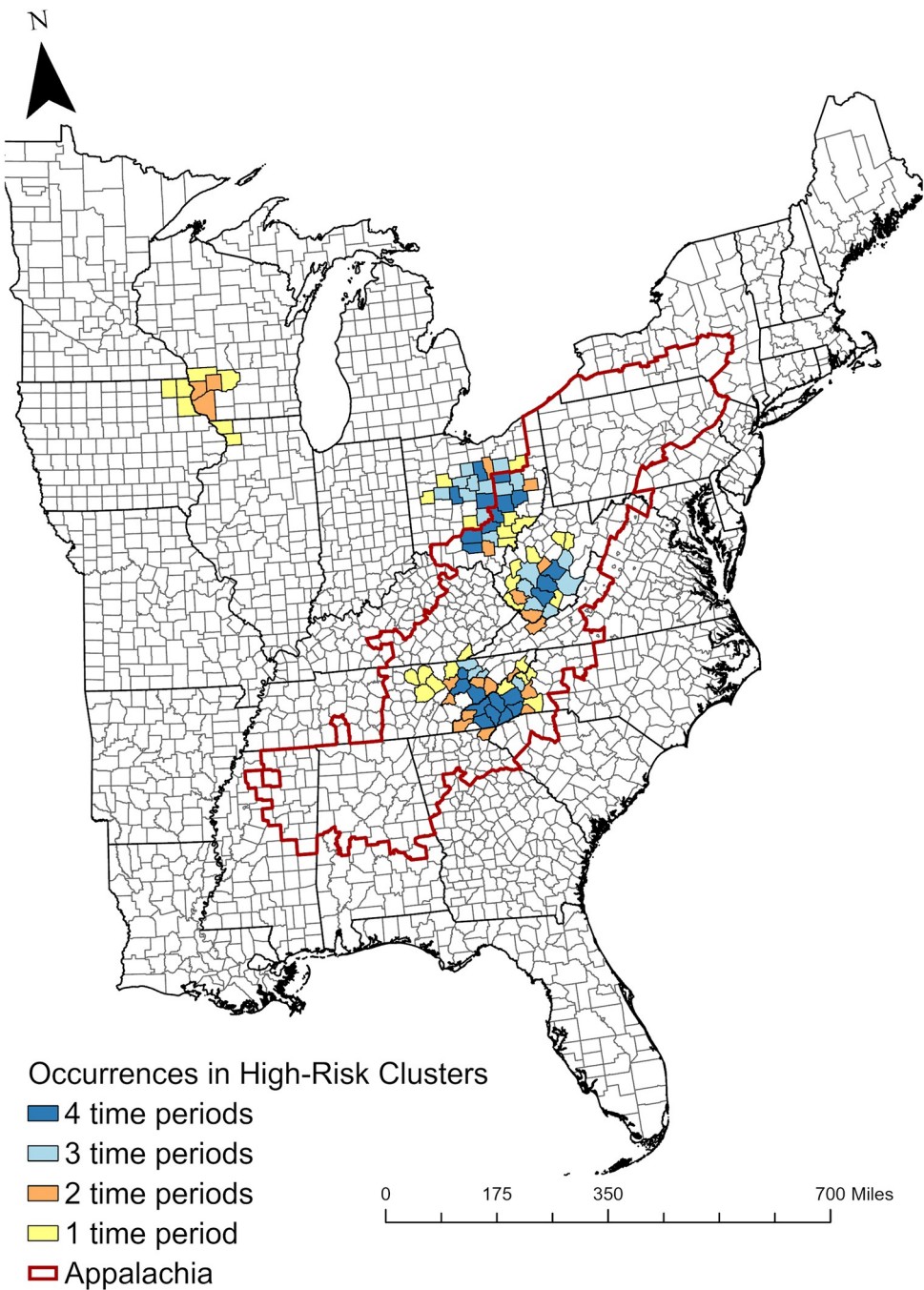

**Fig 4. The number of time periods in which each county fell within a high-risk cluster of La Crosse virus neuroinvasive disease from 2003–2021 in the eastern United States (data obtained from ArboNET).**

included 91% of reported cases in those states and had a CI of 7.8 cases per 100,000 population —nearly 10-times higher than the combined CI.

## Predictors of county-level LACV disease risk

After analyzing the collinearity of predictors, we removed percentage of the population with a high school population or less (correlated with median household income; $r_s$ = -0.74) and

**Table 3. Significant clusters of La Crosse virus neuroinvasive disease detected with Tango's flexibly shaped scan statistic for 2003–2007, 2008–2012, 2013–2017, and 2018–2021.**

| Time Period | Cluster (States[1]) | Cases | # Counties | Total Pop. | CI[2] per 100,000 Pop. | RR[3] |
|---|---|---|---|---|---|---|
| 2003–2007 | 1 (TN, NC, GA) | 115 | 18 | 1,226,677 | 9.4 | 44.9 |
| | 2 (WV, VA) | 91 | 13 | 619,553 | 14.7 | 70.3 |
| | 3 (OH) | 49 | 18 | 1,302,139 | 3.76 | 18.0 |
| | 4 (WI, IA, IL) | 18 | 9 | 204,469 | 8.8 | 42.2 |
| | 5 (TN) | 7 | 3 | 88,409 | 7.9 | 37.9 |
| 2008–2012 | 1 (TN, NC, KY, SC) | 124 | 19 | 1,357,353 | 9.1 | 53.6 |
| | 2 (WV) | 61 | 15 | 625,896 | 9.8 | 57.2 |
| | 3 (OH) | 41 | 14 | 944,049 | 4.3 | 25.5 |
| | 4 (OH) | 15 | 6 | 373,384 | 4.0 | 23.6 |
| 2013–2017 | 1 (TN, NC) | 121 | 23 | 1,556,749 | 7.8 | 53.6 |
| | 2 (OH) | 59 | 23 | 1,469,994 | 4.0 | 27.7 |
| | 3 (WV, VA) | 23 | 11 | 550,196 | 4.2 | 28.8 |
| | 4 (WI) | 9 | 4 | 148,710 | 6.1 | 41.7 |
| 2018–2021 | 1 (TN, NC) | 78 | 20 | 1,511,006 | 5.2 | 50.2 |
| | 2 (OH) | 71 | 26 | 1,999,674 | 3.6 | 34.5 |
| | 3 (WV) | 13 | 6 | 233,496 | 5.6 | 54.2 |

[1]See Fig 1 for state abbreviations

[2] Cumulative incidence

[3] Risk ratio

**Table 4. Results of univariable analyses for potential predictors of county-level LACV-ND risk in Ohio, West Virginia, North Carolina, and Tennessee from 2003–2021.**

| Predictor | Risk Ratio | 95% Conf. Int.[1] | p |
|---|---|---|---|
| % Population under 20 years | 0.84 | 0.73, 0.97 | **0.01** |
| % Population that is White | 1.08 | 1.06, 1.11 | **< 0.001** |
| Median Year Housing Built | 0.95 | 0.92, 0.98 | **< 0.001** |
| Median Household Income per 1,000 USD | 0.97 | 0.94, 0.99 | **0.003** |
| Population Density (km$^2$) per 100 Population | 0.75 | 0.62, 0.90 | **0.002** |

[1]95% Confidence Interval of the risk ratio

**Table 5. Significant predictors of county-level LACV-ND risk in Ohio, West Virginia, North Carolina, and Tennessee from 2003–2021.**

| Predictor | Risk Ratio | 95% Conf. Int.[1] | p |
|---|---|---|---|
| Intercept | 1.16 | -20.32, -16.22 | **< 0.001** |
| % Population that is White | 1.08 | 1.06, 1.1 | **< 0.001** |
| Median Household Income per 1,000 USD | 0.97 | 0.96, 0.99 | **< 0.001** |

[1]95% Confidence Interval of the risk ratio

housing density (correlated with population density; $r_s$ = 0.99) from subsequent analyses. All remaining predictors had significant associations with LACV-ND risk in univariable analyses (Table 4). We retained two predictors after manual backwards elimination: the percentage of the population that was white and median household income. In the final model, risk of LACV-ND tended to be higher in counties where more of the population was white and where the median household income was lower (Table 5). While holding all other variables constant, a one-percent increase in the percentage of the population that was white was associated with an 8% increase in CI of LACV-ND (RR = 1.08) and a $1,000 increase in median household income was associated with a 3% decrease in the CI of LACV-ND (RR = 0.97). There was no evidence of spatial autocorrelation in the residuals of the final model (Moran's I = -0.02; p = 0.93).

## Discussion

Our study demonstrates that the geographic distribution of LACV-ND was consistent from 2003 to 2021, with persistent high-risk clusters in the Appalachian region. The CI of LACV-ND within those clusters was substantially higher than in the study area overall, demonstrating that the primary burden of LACV-ND consistently fell on populations in a few high-risk foci. State and local public health officials should consider the consistent distribution of clusters and allocate resources for targeted interventions to reduce LACV-ND risk.

This study adds to existing evidence that focal risk remains high over extended time periods in places where LACV-ND is reported. For example, a geographic analysis of LACV-ND in Illinois found a consistent pattern of clustering within three counties from 1966–1995, with most cases in the state reported from a single town and the surrounding area [27]. More recently, three unique instances of spatially-linked noncoincident cases of LACV disease were reported in North Carolina (NC) from 2002–2017, with two cases occurring at a single household seven years apart [28]. There are also several examples of LACV-infected mosquitoes being collected at case sites in years following LACV-ND diagnoses [29–31]. Additional research should investigate persistent fine-scale clustering patterns (e.g., among home addresses, census block groups, or census tracts) within the high-risk foci that we identified in this study to better inform targeted public health interventions.

We saw a retraction in the size of geographic clusters throughout the time period that corresponded with an overall reduction in CI. The most conspicuous reduction was in the upper Midwestern region, which once reported the most cases in the US [6], but reported very few cases in recent years. The cause of declining LACV-ND CI is not clear. We are not aware of any large-scale efforts to reduce LACV transmission, but local efforts may have reduced incidence in some areas [32]. It is also possible that risk of LACV-ND has declined in high-risk foci because of increased human immunity or the development of LACV-resistance in mosquito populations [33–35]. Another important consideration is that we could not adjust for age or sex in our calculations of CI; the median age is increasing in Appalachia and the US overall [36], which could confound temporal comparisons of raw CI for LACV-ND, which predominately occurs in children.

While limiting our investigation to states with persistent LACV-ND clusters (Tennessee, North Carolina, West Virginia, and Ohio), we found that cumulative incidence tended to be higher when median household income was low and when more of the population was white. The relationships between those variables and population-level risk appear to represent the underlying structure of the at-risk populations; most cases were reported among counties in Appalachia, a region that is largely comprised of rural, socioeconomically-disadvantaged populations that face severe disparities in disease and mortality risk [36–39]. Because our predictor

investigation did not include the entire study area, these results cannot be generalized beyond the included states. Our model also fails to consider environmental variables that are associated with mosquito-borne disease transmission, which limits the utility of the model for prediction and prevents it from identifying the drivers of the disease distribution.

The principal vector and wildlife hosts of LACV are broadly distributed throughout the eastern US and southeastern Canada [3,4,40], yet most LACV-ND continues to be reported from a small subset of their overlapping ranges in the eastern US. This includes states with persistent LACV-ND like North Carolina and Tennessee, where cases are mostly reported from only a few counties, even though the vectors and hosts are more widely distributed throughout the states. The factors controlling the LACV-ND distribution are unknown, but are likely due to complex relationships between environment and vector interactions [41]. Another consideration is that there are multiple LACV variants, of which only one is known to be pathogenic. Novel LACV variants are sometimes detected outside of high-risk areas, like in New England, where no case of LACV disease has been reported [42]. Environmental characteristics in high-risk LACV-ND foci may be more suitable for the pathogenic LACV variant than others, but research on this subject is deficient.

There are several additional limitations to our study. ArboNET is a passive surveillance system, and its data is limited to cases that are both detected and reported. Cases are reported by county of residence, so travel-associated cases cannot be controlled for and may bias the reported distribution of disease. Clinical detection is another important consideration; La Crosse virus infections are usually asymptomatic, and symptomatic cases can be mistaken as herpes simplex encephalitis or bacterial meningitis [9,11]. Consequently, the distribution of reported LACV-ND could be biased by regional differences in clinical awareness. Another limitation is that the public cannot receive case-specific data at the county level from ArboNET, so we were unable to standardize the cases by age or sex. Therefore, unstandardized demographic structure may confound risk comparisons among geographic areas and time periods. This also prevented us from focusing our study on the people less than 18 years of age, which suffer >90% of LACV-ND cases [5]. Therefore, our calculations underestimate the risk for the most vulnerable group [24,43]. Our predictor investigation was also limited by reliance on previous evidence to identify potential predictors, and it is possible that we failed to identify some significant socioeconomic and demographic predictors.

## Conclusions

The geographic distribution of La Crosse virus neuroinvasive disease (LACV-ND) remained consistent from 2003–2021, with the geographic clusters consistently present in four areas in the Appalachian region: Ohio, West Virginia, east Tennessee, and western North Carolina. County-level risk for LACV-ND in those states tended to be higher when more of the population was white and when median household income was lower. These findings add to the existing body of research indicating that LACV-ND risk is spatially persistent. The consistent distribution of LACV-ND warrants targeted public health interventions to reduce disease incidence in high-risk areas.

## Author Contributions

**Conceptualization:** Corey Allen Day, Agricola Odoi, Rebecca Trout Fryxell.

**Data curation:** Corey Allen Day.

**Formal analysis:** Corey Allen Day.

**Methodology:** Corey Allen Day, Agricola Odoi.

**Supervision:** Agricola Odoi, Rebecca Trout Fryxell.

**Validation:** Corey Allen Day.

**Visualization:** Corey Allen Day.

**Writing – original draft:** Corey Allen Day.

**Writing – review & editing:** Corey Allen Day, Agricola Odoi, Rebecca Trout Fryxell.

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
