## [Decision Letter · Decision Letter 0]

23 Sep 2022

Dear Mr. Day,

Thank you very much for submitting your manuscript "Persistent high-risk geographic clusters of neuroinvasive La Crosse virus disease in the socioeconomically disadvantaged Appalachian region of the United States" for consideration at PLOS Neglected Tropical Diseases. As with all papers reviewed by the journal, your manuscript was reviewed by members of the editorial board and by several independent reviewers. In light of the reviews (below this email), we would like to invite the resubmission of a significantly-revised version that takes into account the reviewers' comments. 

We cannot make any decision about publication until we have seen the revised manuscript and your response to the reviewers' comments. Your revised manuscript is also likely to be sent to reviewers for further evaluation.

Sincerely,

Rhoel Ramos Dinglasan

Academic Editor

Eugenia Corrales-Aguilar

Section Editor

Reviewer's Responses to Questions

**Key Review Criteria Required for Acceptance?**

**Methods**

-Are the objectives of the study clearly articulated with a clear testable hypothesis stated?

-Is the study design appropriate to address the stated objectives?

-Is the population clearly described and appropriate for the hypothesis being tested?

-Is the sample size sufficient to ensure adequate power to address the hypothesis being tested?

-Were correct statistical analysis used to support conclusions?

-Are there concerns about ethical or regulatory requirements being met?

Reviewer #1: Objectives are clearly stated, study design is appropriate, and there are no ethical concerns with the data

Reviewer #2: The methods were acceptable; however, two more novel computational approaches exist that the reviewer encourages the author to try on their dataset. Given the limited dataset available, it is unknown if these statistical approaches hold merit. However, the author is requested to attempt them. Details can be found in the attached pdf document.

Reviewer #3: Please see attached document for my evaluation of the Methods.

**Results**

-Does the analysis presented match the analysis plan?

-Are the results clearly and completely presented?

-Are the figures (Tables, Images) of sufficient quality for clarity?

Reviewer #1: Analyses are appropriate, results and figures are clearly presented.

Reviewer #2: The study findings are logical and evidence-based. The study is distinct from the current scientific literature.

Reviewer #3: Please see attached document for my evaluation of the Results.

**Conclusions**

-Are the conclusions supported by the data presented?

-Are the limitations of analysis clearly described?

-Do the authors discuss how these data can be helpful to advance our understanding of the topic under study?

-Is public health relevance addressed?

Reviewer #1: Conclusions are appropriate, limitations are noted, and public health relevance is indicated. While noting that Appalachia is largely comprised of rural, socioeconomically disadvantaged populations that face severe disparities in disease and mortality risk, do you have any thoughts as to why those of lower median household income are at higher risk for LACV disease specifically? Is there any reason there would be higher mosquito breeding? In regards to higher risk in counties with a higher population proportion of white persons, could this be related to better access to health care?

Reviewer #2: The conclusions are logical and evidence-based.

Reviewer #3: Please see attached document for my evaluation of the Conclusions.

**Editorial and Data Presentation Modifications?**

Reviewer #1: - Consider reporting data to one decimal point only to allow for easier reading 

- Line 103: It is not clear that “probable” cases require laboratory testing, albeit having criteria for notification that are not as strict as those for confirmed cases. Please consider rewording the statement that “laboratory criteria for a confirmed case are not met, cases are reported as probable” 

- Line 213: no capital needed for virus in “West Nile virus” 

- Line 219: Can abbreviate “relative risks” as previously done

- Line: 261: Perhaps consider a more scientific word than “notoriously” 

- Line 288: Do you mean or “increased” or “was higher” in the sentence “County-level risk for neuroinvasive LACV disease in those states increased when more of the population was white or when median household income was lower”.

Reviewer #2: (No Response)

Reviewer #3: Please see attached document for my comments on data presentation and editorial suggestions.

**Summary and General Comments**

Reviewer #1: This is an interesting, nicely written, and well referenced manuscript, and appropriate limitations are noted. The findings are important in terms of considering public health approaches to control.

Reviewer #2: (No Response)

Reviewer #3: I believe these findings would be valuable to the general body of knowledge regarding La Crosse virus disease. Most of the methods seem sound and the use of maps for data visualization is really effective. However, there are some methods and comparisons that I strongly disagree with and the writing is not clear and concise (see attached document for details).

PLOS authors have the option to publish the peer review history of their article (what does this mean?). If published, this will include your full peer review and any attached files.

Reviewer #1: No

Reviewer #2: No

Reviewer #3: No
---

## [Decision Letter · Decision Letter 1]

1 Jan 2023

Dear Mr. Day,

We are pleased to inform you that your manuscript 'Geographically persistent clusters of La Crosse virus disease in the Appalachian region of the United States from 2003 to 2021' has been provisionally accepted for publication in PLOS Neglected Tropical Diseases.

Best regards,

Rhoel Ramos Dinglasan

Academic Editor

Eugenia Corrales-Aguilar

Section Editor

Reviewer's Responses to Questions

**Key Review Criteria Required for Acceptance?**

**Methods**

-Are the objectives of the study clearly articulated with a clear testable hypothesis stated?

-Is the study design appropriate to address the stated objectives?

-Is the population clearly described and appropriate for the hypothesis being tested?

-Is the sample size sufficient to ensure adequate power to address the hypothesis being tested?

-Were correct statistical analysis used to support conclusions?

-Are there concerns about ethical or regulatory requirements being met?

Reviewer #2: Yes

Reviewer #3: (No Response)

**Results**

-Does the analysis presented match the analysis plan?

-Are the results clearly and completely presented?

-Are the figures (Tables, Images) of sufficient quality for clarity?

Reviewer #2: Yes

Reviewer #3: (No Response)

**Conclusions**

-Are the conclusions supported by the data presented?

-Are the limitations of analysis clearly described?

-Do the authors discuss how these data can be helpful to advance our understanding of the topic under study?

-Is public health relevance addressed?

Reviewer #2: Yes

Reviewer #3: (No Response)

**Editorial and Data Presentation Modifications?**

Reviewer #2: Accept

Reviewer #3: (No Response)

**Summary and General Comments**

Reviewer #2: None.

Reviewer #3: I thank the authors for their thoughtful responses and significant revisions. The manuscript now communicates their findings much more clearly and effectively.

PLOS authors have the option to publish the peer review history of their article (what does this mean?). If published, this will include your full peer review and any attached files.

Reviewer #2: No

Reviewer #3: No

---

## [Editor Report · Acceptance letter]

15 Jan 2023

Dear Mr. Day,

We are delighted to inform you that your manuscript, "Geographically persistent clusters of La Crosse virus disease in the Appalachian region of the United States from 2003 to 2021," has been formally accepted for publication in PLOS Neglected Tropical Diseases.

Best regards,

Shaden Kamhawi

co-Editor-in-Chief

Paul Brindley

co-Editor-in-Chief
